# Primary Adenocarcinoma of the Upper Urinary Tract: A Systematic Review of the Literature on a Rare Entity

**DOI:** 10.3390/jcm14062010

**Published:** 2025-03-16

**Authors:** Ilias Giannakodimos, Evripidis Bekiaris, Afroditi Ziogou, Alexios Giannakodimos, Evangelia Mitakidi, Konstantina Psalla, Evangelos Fragkiadis, Aris Kaltsas, Zisis Kratiras, Michael Chrisofos

**Affiliations:** 1Third Department of Urology, Attikon University Hospital, School of Medicine, National and Kapodistrian University of Athens, 12462 Athens, Greece; ares-kaltsas@hotmail.com (A.K.); kratiras.urology@gmail.com (Z.K.); mchrysof@med.uoa.gr (M.C.); 2Department of Urology, Geniko Kratiko Nikaias General Hospital, 18454 Athens, Greece; evripidis.bekiaris@gmail.com; 3Department of Medical Oncology, Metaxa Cancer Hospital, 18537 Athens, Greece; aziogou@yahoo.com (A.Z.); alexisgiannak@hotmail.com (A.G.); 4Departement of Anesthesiology, General Hospital of KAT, 14561 Athens, Greece; evangeliamitakidi@gmail.com; 5Second Departement of Surgery, Geniko Kratiko Nikaias General Hospital, 18454 Athens, Greece; konstantinaps19@gmail.com; 6Departement of Urology, National and Kapodistrian University of Athens, Laikon General Hospital, 11527 Athens, Greece; e.fragkiadis@gmail.com

**Keywords:** adenocarcinoma, upper urinary tract, renal pelvis, ureter, kidney

## Abstract

**Background**: Only a few published cases of primary adenocarcinoma of the upper urinary tract have been described in the literature. The aim of this systematic review was to collect all published cases of primary adenocarcinoma of the upper UT and identify prognostic factors and useful diagnostic modalities for their optimal treatment. **Methods**: Systematic research in the PubMed/Medline and Scopus databases concerning primary adenocarcinoma of the upper urinary tract was performed by two independent investigators. A total of 85 studies were included in the review. **Results**: In total, 84 patients were included, consisting of 54 males (64.29%) and 30 females (35.71%). Out of the available data, 24.71% reported a history of lithiasis, 16.47% episodes of pyelonephritis and 10.59% a history of hydronephrosis. Concerning histologic findings on excised tumors, 52.44% of neoplasms were mucinous, 19.51% tubulovilous, 18.29% papillary, 4.88% mixed mucinous–papillary and 4.88% poorly differentiated. Concerning anatomical origin, 62.34% of tumors were found in the renal pelvis, 22.08% in the ureter and 12.99% in both the renal pelvis and the ureter. Surgical treatment was the preferred therapeutic option and was performed in 96.39% of the included patients. In multivariable analysis, a statistically significant relationship was found between a clinical cure and ureter origin of the lesion (OR: 0.17, 95% CI: 0.00–0.22, *p* = 0.002), the presence of an abdominal mass (OR: 0.08, 95% CI: 0.01–0.63, *p*: 0.034) and a poorly differentiated histological type (OR: 0.02, 95% CI: 0.00–0.91). In multivariable time-to-event analysis, the male sex (HR: 0.12, 95% CI: 0.02–1.01, *p*: 0.019) and poorly differentiated histological type (HR: 91.06, 95% CI: 7.31–1134.32, *p*: 0.002) had statistically significant impacts on overall survival. **Conclusions**: Selection of the optimal surgical management, via either nephrectomy or nephroureterectomy, depends on the origin of the primary lesion and represents the mainstay of treatment. A suspicion from the urologist is needed for the identification and optimal treatment of these rare tumors.

## 1. Introduction

Primary adenocarcinoma of the upper urinary tract can present with a range of diagnostic challenges due to its rarity and a broad array of possible differential diagnoses that can complicate its identification. This malignancy is uncommon, and its differential diagnosis may encompass a variety of cancers originating from either genitourinary, such as urothelial cancer, or extra-genitourinary sources [1]. As a result, physicians are usually confronted with difficulties in both the diagnosis and treatment of this rare malignancy [2]. A clear understanding of how to distinguish between primary adenocarcinomas and secondary neoplasms is crucial, as treatment strategies can vary significantly depending on the underlying pathology. Although transitional cell carcinoma and squamous cell carcinomas comprise the majority of neoplasms arising from either the renal pelvis or the ureter, accounting for 85–90% and 10–15% of reported malignancies, respectively, primary adenocarcinoma of the upper urinary tract [3] remains exceptionally rare, and only a few cases of primary adenocarcinoma of the upper UT have been described in the literature [4,5]. More specifically, adenocarcinomas of the upper UT, including the renal pelvis and ureter, account for less than 1% of published cases [6,7,8,9].

Histologically, these tumors can be subclassified into various types, including tubulovilous, mucinous, papillary nonintestinal and poorly differentiated types, depending on the specific morphological features observed in histological analysis [10]. Various risk factors have been suspected for the development of these tumors, including a history of lithiasis, glandular metaplasia and previous episodes of inflammation [11]. The management of adenocarcinomas in the upper UT remains challenging due to the heterogeneity in tumor biology [12]. The differential diagnoses for primary adenocarcinoma of the upper UT may include tumors arising from mammary-like glands or skin adnexal structures [13]. These malignancies mainly present with atypical symptoms and non-characteristic radiological features, and thus their diagnosis remains challenging and mainly relies on a clinical suspicion from the urologist [4]. Primary adenocarcinoma represents an infrequent subtype of urological malignancy, further complicating its diagnosis and management [14]. As a result, a high degree of clinical suspicion from the urologist is essential to making an accurate diagnosis. Due to the scarcity of this entity, no standardized treatment protocol has been proposed, and management options may include partial or radical nephrectomy and nephroureterectomy, which can be performed via open or laparoscopic approaches. Currently published guidelines present controversies regarding the management of malignancies of the upper UT, mainly due to challenges in disease diagnosis and varying therapeutic options [15]. More specifically, guidelines among different organizations fail to provide further recommendations for the management of rare tumors, such as primary adenocarcinoma of the UT, due to the scarcity of published data [15].

This study systematically reviewed the literature concerning primary adenocarcinoma of the upper UT, highlighting useful diagnostic modalities and optimal treatments for these rare neoplasms. By compiling and analyzing the existing data, we hope to offer more precise guidelines that will aid in the diagnosis and management of this challenging and infrequently encountered malignancy.

## 2. Materials and Methods

### 2.1. Search Strategy

This systematic review was performed according to the Preferred Reporting Items for Systematic Reviews and Meta-Analyses guidelines [3]. The protocol of this study is published in the International Prospective Register of Systematic Reviews (PROSPERO) with reference number CRD42024563514. Two investigators (A.Z. and A.G.) independently searched the PubMed/Medline, Scopus and Google Scholar databases for eligible articles reporting on primary adenocarcinomas of the upper urinary tract, renal pelvis and/or ureter, until 5 December 2024. The following terms were used for the search strategy: (Primary Adenocarcinoma OR Primary Mucinous Adenocarcinoma OR Intestinal Metaplasia) AND (Ureter OR Upper Urinary Tract OR Renal Pelvis OR Renal Calyces). Any controversies were resolved with the intervention of a senior investigator (I.G.). A flowchart (PRISMA diagram) summarizing the selection process is included.

### 2.2. Inclusion Criteria

This systematic review included case reports and case series that referred to primary adenocarcinomas located in the renal pelvis or ureter and were written only in the English language.

### 2.3. Exclusion Criteria

Reviews and systematic reviews were excluded. Inaccessible articles, letters to the editor, comments, articles “epub ahead of print” and studies referring to animal reports were excluded from the systematic review. Studies with an unclear diagnosis or insufficient data were also excluded. Moreover, an additional search of the references of the eligible articles was performed in order to assess potential studies following the snowball procedure.

### 2.4. Data Extraction

Three investigators (E.B., N.P. and E.M.) worked independently and extracted information from all studies included in this systematic review, using a predefined template. Any controversies or data extraction inconsistencies were resolved by a senior investigator (I.G.). Data concerning age, tumor location, symptoms and history of lithiasis or inflammation in the included patients were compiled. The researchers also compiled information regarding diagnostic modalities, histological findings, type of surgery and follow-up. Finally, information concerning the occurrence of death, relapse and optimal treatment was collected as well. Data management and analysis were performed using the STATA 15.0 statistical package (Stata Corporation, StataCorpLL, College Station, TX, USA, 2015).

### 2.5. Statistical Analyses

Numerical variables were presented as the mean ± SD (standard deviation) or as the median (25–75% quartiles) if they were skewed. Categorical variables were presented using frequencies and percentages. Patients included in the case series were considered unique case reports in order to estimate variables of interest. Several studies did not report all outcomes of interest and, therefore, relative rates were estimated based on the available data. Patients’ risk factors were associated with two variables of interest: clinical cure, recorded as a cure after treatment, and incidence of death, recorded as the time interval from surgery to death. Univariable and multivariable logistic regressions were performed to investigate any association of a clinical cure (binary variable) with various risk factors. Univariable and multivariable Cox regression analyses, using Cox’s proportional hazards, were performed to investigate any associations between patients’ survival, based on a time-to-event analysis, and various risk factors.

## 3. Results

### 3.1. Literature Search

The literature search retrieved 1250 studies. After duplicate articles’ removal, record screening and the inclusion of studies derived from a snowball procedure, 85 articles, published from 1932 to 2024, met the inclusion criteria and were finally included in this systematic review. A flow diagram of the selection process is depicted in Figure 1.

### 3.2. Countries of Origin of the Included Articles

Concerning countries of origin of all included articles, 50 cases (58.82%) originated from Asian countries, 16 cases (18.82%) from the USA, 12 cases (14.12%) from European countries, 5 cases (5.88%) from Canada and only 1 case each from a South American (1.18%) and African (1.18%) country, respectively (Figure 2).

### 3.3. Patients’ Demographics

In total, the included articles concerned 84 patients, consisting of 54 males (64.29%) and 30 females (35.71%). The mean age of patients was 57.08 ± 14.58 years (mean ± SD), ranging from 10 to 84 years of age.

### 3.4. Risk Factors

Concerning risk factors for adenocarcinoma development, 21 patients (24.71%) reported a history of lithiasis, 14 patients (16.47%) episodes of pyelonephritis and 9 patients (10.59%) a history of hydronephrosis; furthermore, 6 patients (7.06%) were smokers and 1 patient had been diagnosed with cancer of the UT in the past.

### 3.5. Histological Findings

Among the included studies, 52.44% (43 cases) of neoplasms were histologically proved as mucinous, 19.51% (16 cases) as tubulovilous, 18.29% (15 cases) as papillary, 4.88% (4 cases) as mixed mucinous–papillary and 4.88% (4 cases) as poorly differentiated adenocarcinomas.

### 3.6. Anatomical Origins

Concerning anatomical origins, in 48 patients (62.34%), the tumor was found in the renal pelvis; in 17 patients, in the ureter (22.08%); in 10 patients, in both the renal pelvis and ureter (12.99%); and in 2 patients, in the pelvicalyceal system (2.6%).

### 3.7. Clinical Manifestations

The majority of the included patients complained of flank pain (40 patients, 48.19%), while only 7 patients (8.33%) were asymptomatic. Interestingly, 26 patients (31.71%) developed hematuria, 17 patients (20.73%) fever, 19 patients (23.46%) abdominal pain, 17 patients (20.99%) abdominal mass and 7 patients (8.64%) abdominal swelling. Other reported symptoms included nausea (3 patients, 3.7%), vomiting (3 patients, 3.7%) and lower urinary tract symptoms (LUTSs) (10 patients, 12.5%). A detailed clinical symptomatology of the included patients is presented on Table 1. From the available data, the mean duration of symptoms was estimated at 50 ± 14.63 months, the maximum duration at 72 months and only 6 patients presented with an acute onset of symptoms.

### 3.8. Diagnostic Modalities

From the available data, the primary tumor location was found via imaging modalities in 48 patients (57.83%). Concerning the utilized diagnostic modalities, ultrasound (US) of the UT was performed in 31 patients (37.8%), abdominal computed tomography (CT) in 54 patients (65.85%) and abdominal magnetic resonance imaging (MRI) and PET-CT only in 3 patients (3.61%), respectively. More interventional modalities involved the performance of cystoscopy in 28 patients (33.73%), intravenous pyelography in 27 patients (31.76%) and antegrade and retrograde pyelography in 5 (5.88%) and 15 patients (17.65%), respectively. All imaging modalities performed for the detection of the primary lesion are presented on Table 2. Other diagnostic modalities consisted of cytologic findings and preoperative biopsy of suspicious lesions. Urine cytological examination was performed in 21.62% (16 patients) and preoperative biopsy in 24.32% (18 patients) of the included patients. Cytological findings were found suspicious for malignancy in 37.50% (6 patients) and preoperative biopsy in 83.33% (15 patients) of the performed cases. However, despite imaging and diagnostic examination, the initial diagnosis was missed in 44.74% (34 patients) of the included cases.

### 3.9. Treatment and Outcomes

Surgical treatment comprised the optimal therapeutic option and, in the available data, it was performed for 96.39% (80 patients) of the included patients. Concerning the optimal surgical approach, open surgery was conducted in the majority of cases (91.14%, 72 patients), while a laparoscopic approach was taken only in 8.86% (7 patients) of the reported cases. Of note, initial percutaneous nephrostomy or puncture was performed in 15 patients (18.07%). In the available data, the administration of adjuvant radiotherapy or chemotherapy was reported in 4 (5.26%) and 10 patients (13.16%), respectively. After treatment, a clinical cure was reported in 70% (49 patients) of the treated patients.

### 3.10. Univariable and Multivariable Analysis Key Findings

In univariable analysis, logistic regression models were used to evaluate the associations between a clinical cure and various risk factors. This analysis provided odds ratios (ORs) with 95% confidence intervals (CIs) to estimate the likelihood of achieving a clinical cure based on different factors. Only primary lesions originating from the ureter (OR: 0.25, 95% CI: 0.08–0.79, *p* = 0.019) exhibited a statistically significant association with a clinical cure. However, the age, sex, histological type of tumor, presence of hematuria, abdominal mass and presence of the lesion in CT had no impact on the clinical cure. Multivariable logistic regression models were then applied to control for potential confounders and provide a more accurate estimation of the effect of each risk factor. The analysis revealed statistically significant associations between a clinical cure and ureter origin of the lesion (OR: 0.17, 95% CI: 0.00–0.22, *p* = 0.002) as well as the presence of an abdominal mass (OR: 0.08, 95% CI: 0.01–0.63, *p*: 0.034) and poorly differentiated histological type (OR: 0.02, 95% CI: 0.00–0.91). The results of univariable and multivariable analyses concerning a clinical cure are shown in Table 3.

Patients’ survival was reported in 55 cases and the mean duration of follow-up was estimated at 16.34 months, ranging from 1 month to 8 years. In total, out of the available data, 20 deaths (30.77%) were reported. A total of 25% of deaths were reported at 12 months, while the 6-month and 12-month cumulative survival rates were 18.05% (95% CI: 9.7–35.61%) and 25.20% (95% CI: 14.55–43.62%), respectively.

For survival analysis, Cox proportional hazard models were used to evaluate the impacts of various risk factors on overall survival. This model estimates the hazard ratios [3] with 95% confidence intervals, indicating the risk of death associated with each factor over time. The proportional hazards assumption was verified and met for this analysis.

In univariable analysis, origins of the lesion from both the renal pelvis and ureter (HR: 3.52, 95% CI: 0.71–17.35, *p*: 0.122), a poorly differentiated histological type (HR: 18.55, 95% CI: 3.95–87.04, *p*: 0.000) and the presence of the lesion in CT (HR: 0.20, 95% CI: 0.07–0.57, *p*: 0.002) had a significant impact on patients’ overall survival, while age, sex, hematuria and the presence of an abdominal mass had no significant effect on survival. In multivariable analysis, males (HR: 0.12, 95% CI: 0.02–1.01, *p*: 0.019) and a poorly differentiated histological type (HR: 91.06, 95% CI: 7.31–1134.32, *p*: 0.002) presented a statistically significant impact on overall survival. Survival curves based on different histological findings are analyzed in Appendix A. The mean variance inflation factor (VIF) of the Cox proportional regression analysis was estimated at 1.21. The results of the univariable and multivariable analyses concerning patients’ survival are depicted on Table 4.

### 3.11. Sensitivity Analysis

In the 47 studies published prior to 2000, US of the UT was performed in 20 patients (42.6%), abdominal CT in 42 patients (89.4%), cystoscopy in 9 patients (19.5%), cytology in 8 patients (17%), intravenous pyelography in 9 patients (19.15%) and antegrade and retrograde pyelography in 1 (2.13%) and 5 patients (10.64%), respectively. In the 38 studies published after 2000, US of the UT was performed in 11 patients (31.4%), abdominal CT in 12 patients (34.3%), cystoscopy in 19 patients (52.8%), cytology in 8 patients (22.22%), intravenous pyelography in 18 patients (47.37%%) and antegrade and retrograde pyelography in 1 (2.13%) and 4 patients (10.53%), respectively. Abdominal MRI and PET-CT were performed in three patients and were reported only in studies published after 2000. Performance of laparoscopic surgery was also reported only in studies published after 2000 (seven patients, 15.56%).

After including the year of publication as an independent variable in multivariable analysis, the patients included in studies after 2000 presented a non-significant favorable survival benefit (HR: 0.11, 95% CI: 0.01, 1.36). No significant change was observed in the impact of other factors on patients’ survival in multivariable analysis.

Concerning survival analysis in studies published after 2000, Cox proportional hazard models were used to evaluate the impacts of various risk factors on overall survival after excluding older studies (before 2000). No further statistical analysis could be performed due to the scarcity of events and insufficient number of included cases.

## 4. Discussion

Primary adenocarcinoma of the upper urinary tract constitutes an infrequent entity and only a few case reports have been published in the literature. In a series of 3435 cases of resected renal carcinomas reported by Lai et al., only 2 cases were diagnosed with mucinous adenocarcinoma originated from the urinary tract [11]. To the best of our knowledge, this is the first systematic review collecting the published studies concerning adenocarcinoma of the upper UT. Due to its atypical clinical manifestations and non-specific imaging modalities, urologists should be aware of the existence of this rare entity, its optimal management and prognostic indicators that are related with unfavorable outcomes [16].

To overcome the challenges of non-specific clinical symptoms and imaging findings, a multi-modal diagnostic approach is recommended, utilizing advanced techniques like contrast-enhanced CT scans and MRI [17,18]. In our study, abdominal US (37.8%) and CT (65.85%) were utilized in the majority of cases, while intravenous pyelography (31.76%) and retrograde pyelography (17.65%) comprised other supplementary diagnostic modalities. In the majority of cases, abdominal US presents either with signs of hydronephrosis of the affected kidney or with the identification of a suspicious renal mass [16]. Retrograde urography usually demonstrates a filling defect in the renal pelvis or ureter suggestive of a suspicious lesion, while a CT scan, preferably with contrast enhancement, can identify the tumor along with its formation, extent, local spread and possible metastatic lesions [17]. Despite the progression of diagnostic technology, imaging findings are not specific for this rare entity, the identification of these tumors remains challenging and final diagnosis is usually achieved through postoperative biopsy [4]. However, in our study, urine cytology and preoperative biopsy were performed in 21.62% and 24.32% of patients, respectively. Interestingly, cytological findings were suggestive of malignancy in 37.50% and preoperative histologic findings in 83.33% of the included cases. Although these examinations are indicative of suspicious findings in the UT, they are rarely performed and the ultimate diagnosis is made after excision of the suspicious mass. Finally, according to our systematic analysis, in recently published articles (after 2000), less interventional diagnostic modalities, such as CT, MRI and PET, were more frequently performed; on the contrary, more interventional modalities, such as cystoscopy and intravenous pyelography, were performed in previously published articles (before 2000).

Histologically, three subcategories of primary adenocarcinoma of the UT have been described: tubulovilous, mucinous and papillary nonintestinal adenocarcinoma of the renal pelvis [17]. The two initial classifications, indicative of intestinal adenocarcinoma, account for 93% of occurrences [18]. More specifically, tumors with the villous adenoma subtype comprise the most prevalent histological feature in adenocarcinomas of the renal pelvis, supporting the hypothesis that intestinal-type villous adenoma serves as a precursor to renal pelvis adenocarcinoma. In our systematic review, 52.44% (43 cases) of neoplasms were histologically proved as mucinous adenocarcinomas and comprised the predominant histological subtype (52.44% of included cases), followed by tubulovilous (19.51%), papillary (18.29%) and mixed mucinous–papillary adenocarcinomas (4.88%), while poorly differentiated adenocarcinoma was reported only in 4 cases.

In our analysis, both a clinical cure after treatment and patients’ survival were examined independently and were associated with various factors. More specifically, the patients’ survival was analyzed using two distinct approaches. The first approach treated survival as an event, applying logistic regression to assess the likelihood of survival outcomes based on various factors. The second approach considered survival as a time-dependent event, incorporating the actual duration of survival into the analysis to provide a more detailed understanding of how different variables affect long-term outcomes. By using both methods, this study aimed to offer a comprehensive evaluation of survival patterns and identify key factors that influence patient prognosis. Interestingly, a ureteral origin of the lesion and presence of an abdominal mass during physical examination were both associated with worse clinical outcomes after surgery. This possibly results from the increased difficulty of surgical procedures associated with the anatomical origin and advanced disease. Furthermore, the female sex and poorly differentiated tumor type were related with worse overall survival. Interestingly, poor tumor differentiation constitutes an unfavorable prognostic indicator for both immediate treatment outcomes and time-dependent survival. Compared to other histologies of the upper UT, the poor prognosis of urothelial tumors has been linked to various tumor-related factors such as staging, grading location and multifocality, and patient-related factors, such as baseline chronic kidney disease, diabetes mellitus, body mass index and systemic inflammation [19]. More specifically, in a retrospective study conducted by Zappia et al., nodal staging comprised an independent predictor of a poor prognosis, while no difference in survival rate was reported among different racial groups [20].

Although the exact pathophysiological mechanism of the development of upper UT adenocarcinoma is not well-described, several predisposing factors are considered to contribute to the development of this rare neoplasm [8]. Chronic inflammatory irritation resulting from conditions like urolithiasis, pyelonephritis and hydronephrosis can lead to the proliferation of the urinary epithelium and subsequent malignancy [1,21]. Conditions such as multiple sclerosis, use of indwelling catheters and neurogenic detrusor contractions with high detrusor pressure also generate risks for upper urinary tract damage [22]. Additionally, the pathogenesis may involve the implantation of tumor cells from the upper urinary tract into the bladder, suggesting a potential route for disease progression and metastasis [23]. In the present systematic analysis, 24.71% of the included patients reported a history of lithiasis, 16.47% episodes of pyelonephritis and 10.59% a history of hydronephrosis. Another theory suggests that these tumors may arise from the parenchyma of the kidney, as a result of mistakenly directed or aberrant development during the organ’s formation [11].

Effective diagnosis and optimal management require a multidisciplinary approach, including collaboration between urologists, radiologists and pathologists, as well as the use of molecular genetic testing to differentiate rare tumors from more commonly reported malignancies [13,24]. Surgical treatment remains the mainstay of treatment, with nephrectomy or nephroureterectomy being the optimal approach depending on the lesion’s origin. [4,5,25]. In our systematic review, no further analysis concerning the optimal surgical approach or comparison of open and laparoscopic approaches could be made due to the low number of cases subjected to laparoscopic surgical treatment. However, the existing literature suggests that laparoscopic surgery offers several advantages in terms of reduced blood loss, shorter hospital stays and rapid recovery times compared to open procedures [24]. Additionally, the role of adjuvant therapies, such as radiation or chemotherapy, remains uncertain due to limited data; however, several studies indicate potential benefits in specific cases with disseminated metastatic spread, aggressive histological types or poor prognostic indicators [4,24,26]. In our study, the administration of adjuvant radiotherapy or chemotherapy was reported only in 4 (5.26%) and 10 patients (13.16%), respectively, while no further analysis could be implemented due to missing data. However, according to the published studies, a clinical cure and patients’ overall prognosis are considered satisfying with surgery alone, without any adjuvant remedy. Thus, adjuvant treatment should be considered in more aggressive cases with increased metastatic spread, an aggressive histological type or poor prognostic indicators.

The advancement of molecular technology has led to genomic and molecular profiling of UTUC by identifying various genetic alterations, such as mutations in FGFR3, TERT promoter and chromatin remodeling genes, which have potential implications for both diagnosis and treatment [27]. However, due to the scarcity of primary adenocarcinoma of the UUT, no specific molecular biomarkers have been identified. In our systematic analysis, none of the included studies investigated potential genetic or molecular alterations. As a result, further research is needed at the molecular level, in order to identify specific biomarkers that could aid the diagnostic approach and targeted treatment for this extremely rare malignancy.

To the best of our knowledge, this is the first systematic review of the literature concerning the epidemiology, clinical appearance, diagnostic approach and therapeutic management of primary adenocarcinoma of the upper UT. However, our systematic analysis is subject to certain limitations. Our study included only case reports and case series, which are inherently prone to selection and publication bias, with sufficient data whose credibility mainly depends on the accurate record keeping of each institution. In addition, heterogeneity among institutions concerning surgical approaches and record keeping definitely affects outcomes and time-to-event analysis. The lack of uniform reporting standards primarily affects reliability and limits direct comparison between cases. Future research should focus on well-designed studies with robust patient follow-up to elucidate effective management strategies for these rare neoplasms. Increased awareness among urologists and the development of standardized diagnostic and treatment protocols are recommended as essential for the immediate diagnosis of this rare entity and improvement of patients’ outcomes.

## 5. Conclusions

Primary adenocarcinoma of the upper UT, involving the renal pelvis and/or ureter, constitutes an extremely rare entity, with only a few cases reported in the current literature. Due to the ease of confusing its clinical manifestations and imaging findings with other entities, a suspicion from the urologist is needed for the identification and optimal management of these tumors. Multidisciplinary approaches involving urologists, radiologists, pathologists and oncologists remain crucial for the effective management of these rare tumors. Regular case reviews and interdisciplinary collaboration can significantly enhance the diagnostic accuracy and treatment outcomes. Surgical management including nephrectomy or nephroureterectomy depends on the origin of the primary lesion and constitutes the mainstay of treatment. Further well-designed studies with well-reported patient follow-up are needed to elucidate the optimal management of these rare neoplasms.

## Figures and Tables

**Figure 1 jcm-14-02010-f001:**
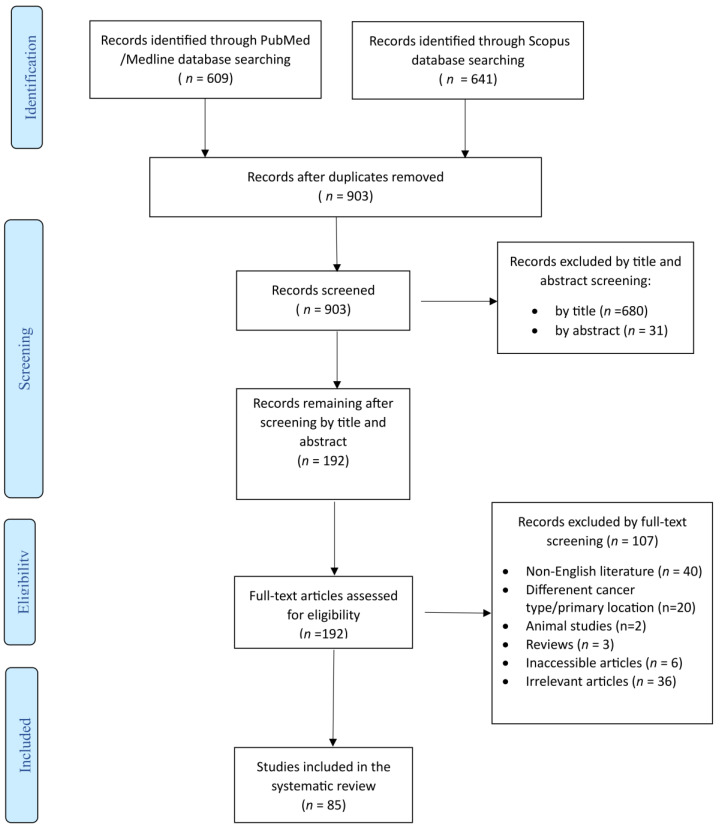
Trial flow of this systematic review and meta-analysis.

**Figure 2 jcm-14-02010-f002:**
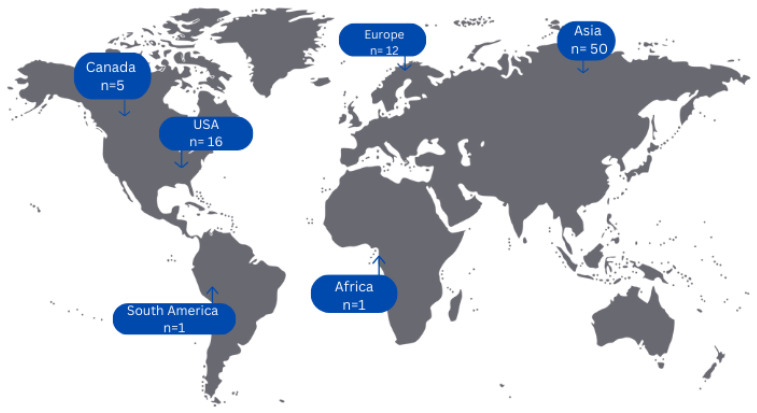
Geographical distribution of cases of primary adenocarcinoma of the upper urinary tract.

**Table 1 jcm-14-02010-t001:** Clinical manifestations of mucinous adenocarcinoma of the upper urinary tract.

Symptoms	Patients (*n* = 85)	Percentages
Flank pain	40	48.19%
Hematuria	26	31.71%
Fever	17	20.73%
Abdominal pain	19	23.46%
Abdominal mass	17	20.99%
Abdominal swelling	7	8.64%
LUTS	10	12.5%
Nausea	3	3.7%
Vomiting	3	3.7%
Asymptomatic	7	8.33%

**Table 2 jcm-14-02010-t002:** Diagnostic modalities for the detection of primary adenocarcinoma of the upper urinary tract.

Imaging Modalities	Patients (*n* = 85)	Percentages
US	31	37.8%
CT	54	65.85%
MRI	3	3.61%
PET-CT	3	3.61%
Cystoscopy	28	33.73%
Intravenous Pyelography	27	31.76%
Antegrade Pyelography	5	5.88%
Retrograde Pyelography	15	17.65%

US: Ultrasonography, MRI: Magnetic Resonance Imaging, CT: Computed Tomography, PET-CT: Positron Emission Tomography–Computed Tomography.

**Table 3 jcm-14-02010-t003:** Risk factors associated with a clinical cure in cases of adenocarcinoma of the upper UT, presenting in univariable and multivariable logistic regression analyses.

Risk Factors	Univariable Analysis			Multivariable Analysis	
		OR (95% CI)	*p*-Value	OR (95% CI)	*p*-Value
Age (years)		0.98 (0.95, 1.02)	0.228	1.00 (0.94, 1.06)	1.06
Sex	Females	Ref		-	
	Males	0.86 (0.29, 2.53)	0.813	0.85 (0.15, 4.91)	0.853
Histological Type	Mucinous	-	-	-	-
	Tubulovilous	1.26 (0.33, 4.86)	0.737	2.96 (0.32, 27.74)	0.342
	Papillary	2.06 (0.38, 11.18)	0.401	1.67 (0.11, 26.35)	0.716
	Poorly Differentiated	0.15 (0.01, 1.64)	0.121	0.02 (0.00, 0.91)	0.044
Location	Renal Pelvis	-	-	-	
	Ureter	1.63 (0.42, 6.33)	0.480	0.02 (0.00, 0.22)	0.002
	Renal Pelvis and Ureter	4.42 (0.88, 22.07)	0.070	0.51 (0.04, 5.93)	0.593
Abdominal Mass	No	-	-	-	-
	Yes	0.53 (0.16–1.74)	0.292	0.08 (0.01–0.63)	0.017
Hematuria	No	-	-	-	-
	Yes	1.45 (0.45–4.71)	0.532	3.59 (0.41–31.33)	0.248
Abdominal CT	No	-	0.678	Ref	0.144
	Yes	1.26 (0.43, 3.66)		0.19 (0.02, 1.74)	

**Table 4 jcm-14-02010-t004:** Risk factors associated with death in cases of adenocarcinoma of the upper UT, presenting in univariable and multivariable Cox proportional regression analyses.

Risk Factors	Univariable Analysis			Multivariable Analysis	
		HR (95% CI)	*p*-Value	HR (95% CI)	*p*-Value
Age (years)		1.03 (0.99, 1.07)	0.188	1.03 (0.97, 1.10)	0.259
Sex	Females	-		-	
	Males	0.48 (0.17, 1.35	0.165	0.12 (0.02, 0.71	0.019
Histological type	Mucinous	-	-	-	-
	Tubulovilous	0.29 (0.04, 2.33)	0.246	0.21 (0.02, 2.49)	0.214
	Papillary	0.43 (0.05, 3.44)	0.430	.	.
	Poorly Differentiated	18.55 (3.95–87.04)	0.000	91.06 (7.31, 1134.32)	0.002
Location	Renal Pelvis	-	-	-	-
	Ureter	2.41 (0.7, 8.27)	0.161	10.67 (0.67, 170.69)	0.094
	Renal Pelvis and Ureter	3.52 (0.71, 17.35)	0.070	7.46 (0.32, 173.25)	0.337
Abdominal Mass	No	-	-	-	-
	Yes	1.68 (0.57–4.91)	0.346	0.25 (0.02–3.1)	0.282
Hematuria	No	-	-	-	-
	Yes	0.82 (0.26–2.58)	0.737	0.31 (0.02–5.23)	0.416
Abdominal CT	No	-	0.002	-	0.212
	Yes	0.2 (0.07, 0.56)		0.22 (0.02, 0.39)	

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
