# Peer review of "Primary Adenocarcinoma of the Upper Urinary Tract: A Systematic Review of the Literature on a Rare Entity"

_jcm, 2025, doi:10.3390/jcm14062010_

Round 1
Reviewer 1 Report
Comments and Suggestions for Authors
This systematic review provides a valuable synthesis of available literature on primary adenocarcinoma of the upper urinary tract, highlighting key prognostic factors and treatment options. The study addresses an exceptionally rare malignancy, primary adenocarcinoma of the upper urinary tract (UT), making a valuable contribution to the limited body of literature. However, methodological limitations, data inconsistencies, and lack of novel diagnostic insights reduce its overall impact. Future studies should aim for prospective data collection, genetic analysis, and standardization of diagnostic and treatment protocols to enhance clinical applicability.
Although here are a few remarks and some recommendations to consider increasing the quality of research:
- The study only includes case reports and case series, which are inherently prone to selection and publication bias.
- The study combines cases from multiple institutions, where diagnostic and therapeutic approaches vary significantly.
- The lack of uniform reporting standards affects reliability and limits direct comparison between cases.
- While surgical management is well-covered, adjuvant treatments (chemotherapy, radiotherapy) are underexplored.
- It would be beneficial to include a discussion on molecular markers and targeted therapies for potential future management.
- The study fails to compare laparoscopic vs. open surgery, despite mentioning that laparoscopic surgery offers benefits (e.g., reduced blood loss, shorter hospital stay).
This systematic review addresses an important clinical gap but suffers from heterogeneous data sources, lack of high-quality studies, and limited discussion on adjuvant treatments. The methodology is sound, and the statistical analysis is well-executed, but further prospective studies and molecular investigations are necessary to enhance clinical application.
It is a good but requires higher-level evidence and broader discussion on treatment options.

Author Response
REVIEWER 1
This systematic review provides a valuable synthesis of available literature on primary adenocarcinoma of the upper urinary tract, highlighting key prognostic factors and treatment options. The study addresses an exceptionally rare malignancy, primary adenocarcinoma of the upper urinary tract (UT), making a valuable contribution to the limited body of literature. However, methodological limitations, data inconsistencies, and lack of novel diagnostic insights reduce its overall impact. Future studies should aim for prospective data collection, genetic analysis, and standardization of diagnostic and treatment protocols to enhance clinical applicability.
Thank you for your interesting comments.
Although here are a few remarks and some recommendations to consider increasing the quality of research:
Comment 1: The study only includes case reports and case series, which are inherently prone to selection and publication bias.
Reply: Thank you for your interesting comment. In the discussion section, we had already mentioned that our systematic analyses included only cases reports and case series and this is a limitation of our study. But unfortunately, only case reports or case series have been published in the literature and management of this rare entity should be based on these existing data. However, he following sentence “which are inherently prone to selection and publication bias” was added in the Discussion section.
Comment 2: The study combines cases from multiple institutions, where diagnostic and therapeutic approaches vary significantly. The lack of uniform reporting standards affects reliability and limits direct comparison between cases.
Reply: Thank you for your valuable comments, but this statement has already been included as a limitation of our study in the Discussion section. However, the following sentence was added in the Discussion section as you proposed: “The lack of uniform reporting standards affects reliability and limits direct comparison between cases”
Comment 3: While surgical management is well-covered, adjuvant treatments (chemotherapy, radiotherapy) are underexplored.
Reply: Thank you for your remark. As already mentioned in the results section only few studies used adjuvant treatment. The role of adjuvant treatment in these patients was also explored in the Discussion section from lines 314-323.
Comment 4: It would be beneficial to include a discussion on molecular markers and targeted therapies for potential future management.
Reply: Thank you for your interesting suggestion. The following paragraph was added in the Discussion section: “Advancement … malignancy”
Comment 5: The study fails to compare laparoscopic vs. open surgery, despite mentioning that laparoscopic surgery offers benefits (e.g., reduced blood loss, shorter hospital stay).
Reply: Thank you for your interesting comment. In our systematic analysis, the majority of patients underwent open surgery and only 7 patients laparoscopic surgery. However, no further information concerning blood loss or hospital stay were described in case reports and thus, any further analysis could not be made. This statement was already included in the Discussion section in lines 309-314.
This systematic review addresses an important clinical gap but suffers from heterogeneous data sources, lack of high-quality studies, and limited discussion on adjuvant treatments. The methodology is sound, and the statistical analysis is well-executed, but further prospective studies and molecular investigations are necessary to enhance clinical application. It is a good but requires higher-level evidence and broader discussion on treatment options.
Thank you for your exciting comments.
Reviewer 2 Report
Comments and Suggestions for Authors
The manuscript addresses an underexplored topic by systematically reviewing primary adenocarcinoma of the upper urinary tract, a rare and challenging malignancy. however different aspect should be improved:
- The study protocol should also include more detail about how data extraction inconsistencies were resolved among reviewers.
- While the background is informative, it could be condensed by removing redundant historical context. In the intro mention how current guidelines lack th focus on the topic (doi: 10.3390/cancers16061115.) + focus more on the current knowledge gap.
- Why Embase and Web of Science were excluded from research strategies?
- Given the broad time span of included studies (1932–2024), variability in diagnostic standards, surgical approaches, and follow-up durations could introduce bias. Consider excluding the oldest reports vs sensitivity analysis based on publication date
- Cox regression analyses are commendable, but variance inflation factor (VIF) to confirm no collinearity would be important
- HR of 91.06 for poorly differentiated histology (p=0.002) suggests a dramatic survival impact but lacks context regarding absolute risk or survival curves. Please explain.
- While the discussion contextualizes findings well, it lacks comparison with other rare upper UT malignancies, such as urothelial carcinoma. consider citing doi: 10.1016/j.clgc.2024.102220. + doi: 10.21037/tau-22-882.
Check redundancies and typos
Author Response
REVIEWER 2
The manuscript addresses an underexplored topic by systematically reviewing primary adenocarcinoma of the upper urinary tract, a rare and challenging malignancy. however different aspect should be improved:
Comment 1: The study protocol should also include more detail about how data extraction inconsistencies were resolved among reviewers.
Reply: Thank you for your interesting comment. The following sentence was added in the Data Extraction section: “Any controversies or data extraction inconsistencies were resolved by a senior investigator (I.G.).”
Comment 2: While the background is informative, it could be condensed by removing redundant historical context. In the intro mention how current guidelines lack th focus on the topic (doi: 10.3390/cancers16061115.) + focus more on the current knowledge gap.
Reply: Thank you for your valuable remark. The sentence regarding historical context was erased form the discussion section. Additionally, the following senetence:” Currently published … published data” were added in the Introduction section to declare how current guidelines lack the focus and knowledge about the topic.
Comment 3: Why Embase and Web of Science were excluded from research strategies?
Reply: Thank you for your interesting comment. Unfortunately, the authors did not have access to these platforms and thus these platforms were excluded from research strategies.
Comment 4:Given the broad time span of included studies (1932–2024), variability in diagnostic standards, surgical approaches, and follow-up durations could introduce bias. Consider excluding the oldest reports vs sensitivity analysis based on publication date
Reply: Thank you for your interesting comment. A paragraph of sensitivity analysis was included in the Results section. In addition, the following sentence was added in the Discussion section: “Finally, according to our systematic analysis, in recently published articles (after 2000), less interventional diagnostic modalities, such as CT, MRI, PET, were more frequently performed, while more interventional modalities, such as cystoscopy and intravenous pyelography, were performed in elderly published articles (before 2000).”
Comment 5: Cox regression analyses are commendable, but variance inflation factor (VIF) to confirm no collinearity would be important
Reply: Thank you for your important comment. VIS for the cox regression analysis was estimated and the following sentence was added in the Results section: “Mean variance inflation factor (VIF) of the cox proportion regression analysis was estimated at 1.21”.
Comment 6: HR of 91.06 for poorly differentiated histology (p=0.002) suggests a dramatic survival impact but lacks context regarding absolute risk or survival curves. Please explain.
Reply: Supplementary Image 1 along with the following sentence regarding survival curves based on different histologic findings were added in the Results section: “Survival curves based on different histologic findings are analyzed in Supplementary Figure 1.”.
Comment 7: While the discussion contextualizes findings well, it lacks comparison with other rare upper UT malignancies, such as urothelial carcinoma. consider citing doi: 10.1016/j.clgc.2024.102220. + doi: 10.21037/tau-22-882.
Reply: Thank you for your valuable suggestion. The following sentences: “Compared to… different racial groups” were added in the Discussion section as you proposed.
Round 2
Reviewer 2 Report
Comments and Suggestions for Authors
The manuscript has improved significantly, addressing previous concerns with clarity and depth. The revised version presents a well-structured systematic review, and I am satisfied with the improvements made.
Comments on the Quality of English LanguageThe quality of English is generally good